biomechanics, evolution

testudine, morphology, neck, allometry, ontogeny, biomechanics

**Author for correspondence:**
Jonathan R. Codd
e-mail: jonathan.codd@manchester.ac.uk

# Turning turtle: scaling relationships and self-righting ability in *Chelydra serpentina*

Ilan M. Ruhr[1], Kayleigh A. R. Rose[3], William I. Sellers[2], Dane A. Crossley II[4] and Jonathan R. Codd[1]

[1]School of Biological Sciences, and [2]Department of Earth and Environmental Sciences, University of Manchester, Manchester, UK
[3]Department of Biosciences, Swansea University, Swansea, UK
[4]Department of Biological Sciences, University of North Texas, Denton, TX, USA

IMR, 0000-0001-9243-7055; KARR, 0000-0001-7023-2809; WIS, 0000-0002-2913-5406; DACII, 0000-0001-9683-7013; JRC, 0000-0003-0211-1786

Testudines are susceptible to inversion and self-righting using their necks, limbs or both, to generate enough mechanical force to flip over. We investigated how shell morphology, neck length and self-righting biomechanics scale with body mass during ontogeny in *Chelydra serpentina*, which uses neck-powered self-righting. We found that younger turtles flipped over twice as fast as older individuals. A simple geometric model predicted the relationships of shell shape and self-righting time with body mass. Conversely, neck force, power output and kinetic energy increase with body mass at rates greater than predicted. These findings were correlated with relatively longer necks in younger turtles than would be predicted by geometric similarity. Therefore, younger turtles self-right with lower biomechanical costs than predicted by simple scaling theory. Considering younger turtles are more prone to inverting and their shells offer less protection, faster and less costly self-righting would be advantageous in overcoming the detriments of inversion.

## 1. Introduction

Predator–prey dynamics drive adaptations in animals, including the evolution of protective armour. Body armour takes many forms and is widespread in extant reptiles, where spines, spikes and osteoderms are commonplace [1]. Arguably, one of the most recognizable forms of body armour is the shell of turtles and tortoises, which is comprised of a dorsal carapace and a ventral plastron, features that distinguish them from all other vertebrates. Despite similarities in general appearance among Testudines, shell morphology varies substantially. For example, generally species that frequently swim [2] or burrow [3] have flatter shells that can be flexible, whereas those requiring better protection from predation [4], desiccation [5] and fluctuating body temperature [5] have taller, more rigid shells [6,7]. Testudine shells are dynamic structures and can also have important physiological roles, which include acting as blood-pH buffers and as reservoirs for water, fat or wastes [8]. While shells impact all aspects of testudine biology, it is locomotor performance that is perhaps the most profoundly affected [9].

In almost all tetrapods, a flexible vertebral column is an important contributor to locomotion. However, in Testudines, only the neck and tail are flexible, because the spine is fused with the underside of the dermal plates to form a hard shell [10]. The pectoral and pelvic girdles are also located inside the shell, which restrict movement of the limbs. As a result of this inflexible body, when traversing uneven surfaces, encountering predators or engaging in reproductive combat they are prone to inverting. Turning upside-down can have serious life-and-death consequences. Once flipped onto their backs, Testudines are susceptible to thermal stress, starvation, stranding and predation, if they cannot effectively self-right [11,12]. Indeed, improved

self-righting performance is associated with higher survival [13] and can have a substantial impact on an individual's fitness [14]. Accordingly, understanding the underlying mechanisms and constraints on self-righting has strong ecological relevance. Self-righting is also an intriguing biomechanical behaviour especially in animals that are long-lived and continue to grow throughout their lives [15]. For example, the common snapping turtle (*Chelydra serpentina*) grows from a carapace length of approximately 30 mm and body mass of 10 g, as a hatchling [16], to a carapace length of over 50 cm and weighing over 40 kg, as an adult. Snapping turtles retain a high degree of carapace rotation as they walk [17] and the neck remains the primary driver in self-righting [18], throughout their lives.

The ability to self-right is dependent on body size, body shape and flexibility of the limbs, neck or tail [19–21]. There are two distinct mechanisms by which Testudines self-right: (i) rotating the limbs, to generate rocking movements to ultimately induce body rolling; or (ii) extending the neck, to directly push against the ground and flip the animal over [20,22]. Investigations of self-righting in Testudines are often limited to theoretical models of the impact of shell shape [20,23,24], the time to self-right (e.g. [25]), and biotic or abiotic influences (e.g. [11]). To our knowledge, just one study has looked at the biomechanics of self-righting [19]. The challenge of self-righting is that the inverted animal is in a stable and low-gravitational potential-energy state. To self-right, Testudines must add gravitational potential energy to the system, by rotating the shell, until it reaches a tipping point, from which it will then roll to the non-inverted stable state and, thereby, overcoming the so-called 'potential hill' [20,24]. Theoretically, shells that are very high and domed should be the easiest to self-right, because the required change in height of the centre of mass is relatively small. Conversely, species with flatter shells will need to raise the centre of mass by a greater extent [20]. Testudines that self-right by limb movements often have more domed shells [20], whereas those with flatter shells, such as mud (kinosternids), pond (emydids), snapping (chelydrids) and soft-shelled (trionychids) turtles, create thrust with their necks to self-right [20,22]. For geometrically similar animals, the minimum energy required to self-right should increase with mass$^{4/3}$, since the height change will scale with mass$^{1/3}$ and the potential-energy change is proportional to the change in mass × height. In neck-based self-righting, this energy comes from a single work loop, and, since the work available in a work loop scales approximately isometrically with mass, we would expect neck-based self-righting to become progressively more difficult as body size increases.

Self-righting speed and energetics might be particularly important in smaller juveniles, which are more prone to inversion and possess shells that offer little anti-predator defence. To self-right more quickly, selection could act on shell morphology; individuals with more domed-shaped shells should self-right more quickly and with less effort [23,24], which might predict that shell shape changes during ontogeny. Alternatively, selection could act on the neck, since it is the primary structure these turtles use to self-right [22]. The aim of this study, therefore, was to examine the influence of body mass, shell shape and neck length on neck-powered self-righting ability and the accompanying biomechanical costs, in a freshwater turtle species, *C.*

*serpentina*. We used animals of different ages to provide the required variation in body size. We measured the self-righting neck force (which we used to estimate kinetic energy and power output) to investigate the scaling relationships between the physical effort to self-right and body mass. Although we would expect scaling of mass$^{4/3}$ for the self-righting effort of geometrically similar shell shapes, we predicted that, due to selection against the possible increase in risks associated with being inverted, self-righting should be easier in younger/smaller turtles, which would be reflected in the speed and biomechanical cost.

## 2. Material and methods

### (a) Animals

Non-gravid female common snapping turtles (*C. serpentina*), ranging in body mass (254.3–4515 g; $n = 33$) and age (less than 1–1.5, 4.5 and 5.5 years old, $n = 26$, 4 and 3 turtles, respectively), were selected for the present study (electronic supplementary material, figure S1). Turtles were housed at 26°C, in small groups, within large plastic tubs (1.5 m wide, 1 m tall), with access to shallow water. All experimental trials took place at 26°C.

### (b) Experimental setup and data collection

Before the commencement of any self-righting trials, morphological measurements of carapace width, carapace length and shell height were taken, with digital calipers (Duratool, model D02264, Premier Farnell, Leeds, UK). Neck length was determined by encouraging the turtle to bite a piece of leather, then grasping the turtle's head (while wearing protective gloves) and gently extending it out from the shell to its full length. Using the calipers, the distance from the shell at the base of the neck to the tip of the snout was then measured and used as a proxy for neck length.

The experimental setup consisted of a force plate, with a pressure pad on top, covered by a thin rubber mat. The force plate (3D Force Plate Type 9286B, Kistler Instruments, Hook, Hampshire, UK) was used to measure the vertical reaction force exerted by a turtle during self-righting. Force data were recorded (at 420 Hz), using the BioWare data-acquisition software (type 2812A, Kistler). To measure the relative contributions of the neck and body to the vertical force, a pressure pad (Pressure Mapping Sensor 7101, Tekscan, South Boston, MA, USA) was placed on top of the force plate. The pressure pad data were recorded (at 100 Hz) using the FootMat Research software (v.7.1, Tekscan). A camera (Sony Cyber-shot RX10 III) was used to record videos (at 100 fps) of the self-righting movements. The self-righting times were calculated from videos, using Tracker (https://physlets.org/tracker; The Open Source Physics Project). The self-righting time is defined as the duration from the moment a turtle's head first contacted the pressure pad/force plate until the head was no longer in contact with the setup (see figure 1a–d).

### (c) Data analysis

Vertical force distributions (see figure 1e) were obtained from the pressure-pad data and used to calculate the relative magnitude of neck force production to the total ground reaction force for the turtle, as shown by the representative curves in figure 1f. This relative value was then multiplied by the absolute vertical force that was recorded by the force plate (figure 1f), which allowed us to determine absolute neck force in newtons. All force-plate and pressure-pad data were filtered with a 10 Hz, two-pole

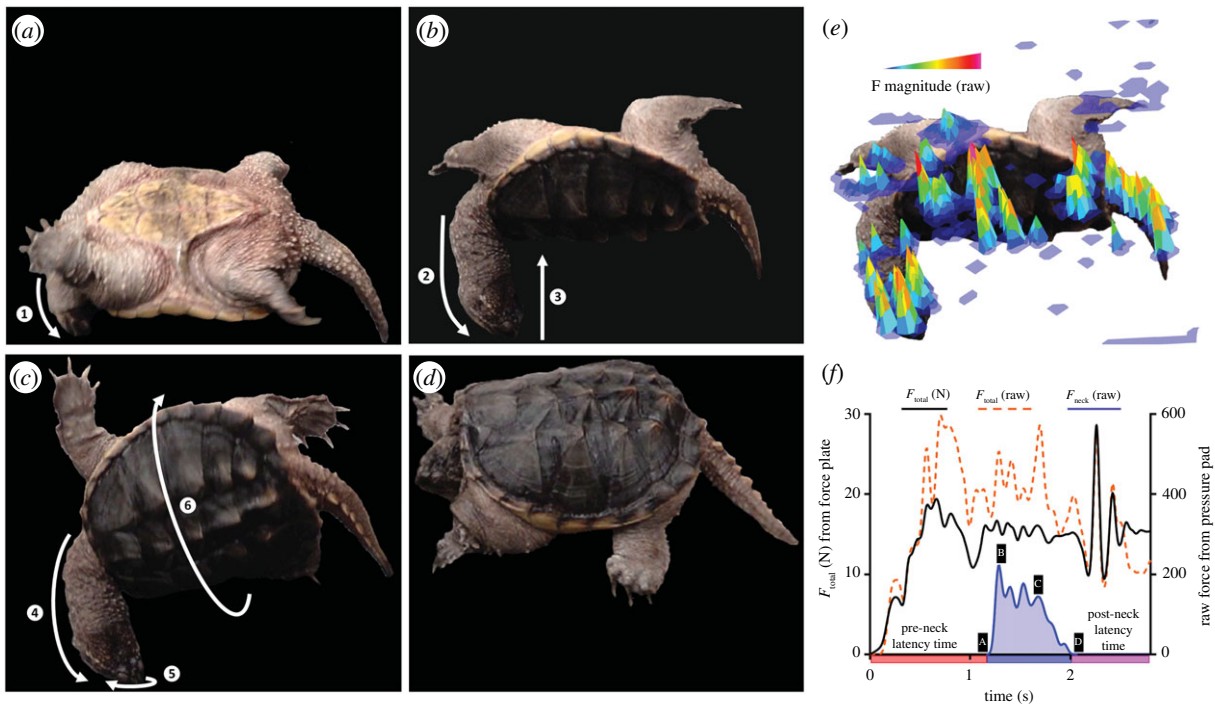

**Figure 1.** Representative images showing a sequence of the progressive stages in a self-righting snapping turtle. (*a*) Initial contact of ① the neck to the ground; (*b*) ② extension of the neck, which ③ begins lifting the shell off the ground; (*c*) ④ full extension, ⑤ twisting of the neck and ⑥ rotation of the shell about its axis; and (*d*) completion of the self-righting manoeuvre, with all four limbs contacting the ground. White lines with arrows represent the direction of motion. The biomechanical effort involved for different body areas is depicted as (*e*) a representative image showing the integrated force peaks during the self-righting manoeuvre depicted in (*b*), highlighting the dynamic distribution of forces across the turtle and the concentration of force application via the neck. The relative magnitude of the force (*F* magnitude) production is indicated by colours, as shown in the panel legend. (*f*) Representative traces of absolute total force ($F_{total}$; black line) in newtons (N) produced by a snapping turtle during one self-righting manoeuvre and recorded by a force plate (indicated in the panel legend). A pressure pad recorded the contribution of neck force ($F_{neck}$; blue line) and total force (red line) exerted by the turtle during self-righting (indicated in the panel legend to the left). The sequence of the self-righting manoeuvre labelled from A to D, to match the corresponding panels are indicated over the blue $F_{neck}$ line. The red, blue and purple rectangular sections found under the *x*-axis correspond to the pre-neck latency—time it took for a turtle to place its head on the ground to initiate the self-righting manoeuvre, self-righting—when force is being applied via the neck, and post-neck latency—the duration of time to complete the self-righting manoeuvre when the turtle no longer used the neck to flip over times that are shown in electronic supplementary material, figure S2 for reference. (Online version in colour.)

Butterworth low-pass filter, using the filtfilt function on Matlab (v.R2020a, MathWorks, Natick, MA, USA), which reduced high-frequency noise present in the data.

Previous reports have shown that shell morphology affects self-righting time [20,23,25]. Therefore, we calculated two indices of shell shape: sphericity index (SI) and flatness index (FI) [23], as defined in equations (2.1) and (2.2).

$$SI = \left(\frac{W \times H}{L^2}\right)^{\frac{1}{3}} \qquad (2.1)$$

and

$$FI = \frac{L + W}{2H}, \qquad (2.2)$$

where $W$ and $L$ are the maximum carapace width and length, respectively, and $H$ is the shell height. Larger sphericity and flatness values indicate greater and flatter shell curvature, respectively.

Impulse ($J$) was calculated as the area under the force–time curve (figure 1*f*), using equation (2.3):

$$J = \sum F_t \Delta t, \qquad (2.3)$$

where $F_t$ is the instantaneous vertical force and $\Delta t$ is the time increment.

From the impulse, kinetic-energy equivalent (KEE) was calculated (assuming a start from rest, so that the initial

momentum is zero), using equation (2.4):

$$KEE = \frac{J^2}{2M_b}, \qquad (2.4)$$

where $M_b$ is the body mass.

From KEE, mean power-output equivalent (PE) was calculated, using equation (2.5):

$$PE = \frac{\Delta KEE}{t_{Flip}}, \qquad (2.5)$$

where $t_{Flip}$ is the self-righting time.

Finally, from KEE, height-change equivalent ($\Delta$HE) was calculated and normalized to carapace width, using equation (2.6):

$$\Delta HE = \frac{\Delta KEE}{M_b \Delta g \Delta W}, \qquad (2.6)$$

where $g$ is the gravity (9.81 m s$^{-2}$).

$\Delta$HE was calculated as a measure of self-righting efficiency, given that we would expect the minimum $\Delta$HE to be half the shell width with a flat shell, and less than half for more rounded shells. It can be higher too if the turtle does not choose the most efficient trajectory, and if the KEE at maximum height is still substantial. Lower $\Delta$HE values thus indicate higher self-righting energetic efficiency.

All data were graphed with GraphPad Prism 8 (GraphPad Software, San Jose, CA, USA). To determine scaling relationships, data were log-transformed and regression lines plotted,

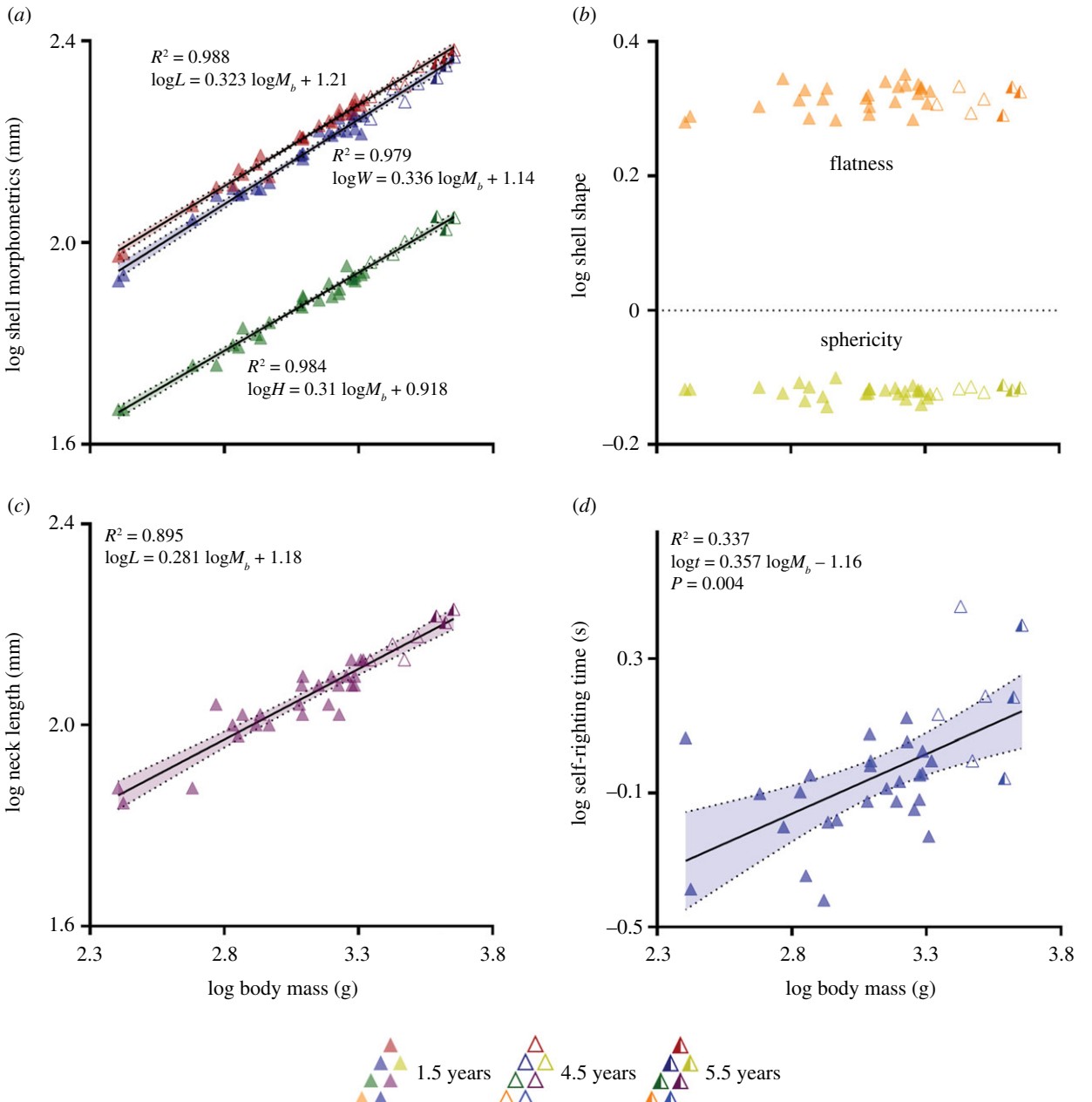

**Figure 2.** Relationships with body mass between (*a*) carapace length (red triangles), carapace width (blue triangles) and shell height (green triangles); (*b*); shell shape (sphericity and flatness indices); (*c*) neck length, during ontogeny; and (*d*) self-righting time, during ontogeny, in female common snapping turtles (*n* = 33). All morphometric parameters positively correlated with body mass. Carapace length, carapace width and shell height followed isometric growth, whereas neck length followed a negative allometric pattern. Self-righting time, during which a snapping turtle uses its head to flip over, is positively correlated with body mass and follows isometric scaling. Simple linear regressions were used to produce best-fit lines through the data. The ages (in years) of snapping turtles used in this study are indicated by closed triangles, for 1.5 y (*n* = 26); open triangles, for 4.5 y (*n* = 4); and half-closed triangles, for 5.5 y (*n* = 3), as shown in the figure legend. Abbreviations: body mass, $M_b$; grams, g; length, *L*; width, *W*; height, *H*; *t*; time. (Online version in colour.)

with the equation $\log(y) = \log(a) + b \cdot \log(x)$, using the ordinary least-squares (OLS) method on GraphPad and coefficients of determination ($R^2$) were calculated. Isometric and allometric scaling relationships were determined by comparing the predicted slope with the allometric slope (*b*), using the 95% confidence intervals (CIs). Assuming geometric similarity (i.e. isometry) across body mass ($M_b$), all linear dimensions were expected to scale to $M_b^{1/3}$; force was expected to scale to $M_b^{2/3}$; KEE should scale as $M_b^{4/3}$; self-righting time should scale as $M_b^{1/2}$; and mean PE as $M_b^{5/6}$. Formal derivations of these predicted relationships are in the electronic supplementary material. Scaling relationships were considered to show isometry when the predicted slope fell within the 95% CIs ($0.95 \leq b < 1.05$), positive allometry when predicted $b > 1.05$, and negative allometry when predicted $b \leq 0.95$. For the derivation of these

scaling relationships, see Derivation of Scaling Predictions in the electronic supplementary material.

## 3. Results

### (a) Morphometrics

The log–log models fit the data extremely well for linear shell dimensions, with $R^2$ values ranging from 0.979 to 0.988 and the 95% CIs of the slopes overlapping the ⅓ value that would be predicted from geometric similarity, thereby, providing no evidence for shell shape change (as defined in the current study) with increasing body mass during ontogeny (figure 2*a*, table 1; electronic supplementary material,

**Table 1.** Scaling relationships between morphometric/biomechanical variables and body mass or between morphometrics and maximum neck force. The regression slope indicates proportional change in variable size with increasing body mass, and 95% confidence intervals (CIs) are shown ($n = 33$). Measured slopes in agreement (using 95% confidence intervals) with predicted slopes from our geometric model are indicated by a check mark ($\checkmark$) and measured slopes lesser or greater than model predictions are indicated by negative signs (−) and positive signs (+), respectively.

| dependent variable | independent variable | slope predicted by geometric model | measured slope | in agreement with model prediction? | lower 95% CI | upper 95% CI | $R^2$ | $p$-value |
|---|---|---|---|---|---|---|---|---|
| carapace length (mm) | body mass (kg) | 0.33 | 0.323 | $\checkmark$ | 0.31 | 0.336 | 0.988 | ≤0.001 |
| carapace width (mm) | body mass (kg) | 0.33 | 0.336 | $\checkmark$ | 0.318 | 0.354 | 0.979 | ≤0.001 |
| carapace height (mm) | body mass (kg) | 0.33 | 0.31 | $\checkmark$ | 0.295 | 0.335 | 0.984 | ≤0.001 |
| neck length (mm) | body mass (kg) | 0.33 | 0.281 | − | 0.246 | 0.316 | 0.895 | ≤0.001 |
| self-righting time (s) | body mass (kg) | 0.5 | 0.357 | $\checkmark$ | 0.173 | 0.54 | 0.337 | 0.004 |
| neck force (N) | body mass (kg) | 0.67 | 0.901 | + | 0.757 | 1.045 | 0.84 | ≤0.001 |
| kinetic-energy equivalent (J) | body mass (kg) | 1.33 | 1.548 | + | 1.341 | 1.755 | 0.882 | ≤0.001 |
| power output (W) | body mass (kg) | 0.83 | 1.191 | + | 0.961 | 1.422 | 0.782 | ≤0.001 |
| neck force (N) | neck length (mm) | 2 | 2.976 | $\checkmark$ | 2.447 | 3.505 | 0.809 | ≤0.001 |
| | carapace length (mm) | 2 | 2.77 | $\checkmark$ | 2.325 | 3.215 | 0.839 | ≤0.001 |
| | carapace width (mm) | 2 | 2.607 | $\checkmark$ | 2.146 | 3.067 | 0.811 | ≤0.001 |
| | shell height (mm) | 2 | 2.837 | $\checkmark$ | 2.34 | 3.334 | 0.814 | ≤0.001 |

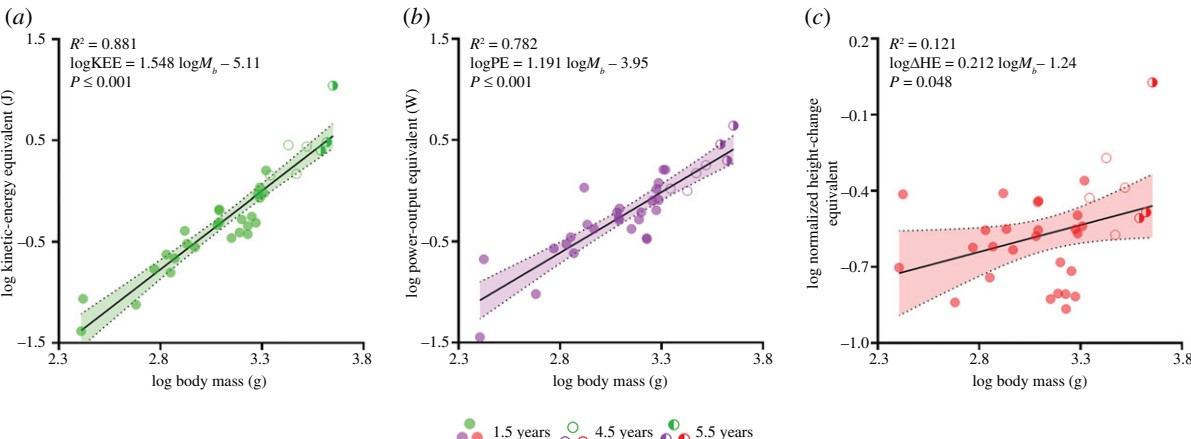

**Figure 3.** Relationship between body mass and (*a*) kinetic-energy equivalent, (*b*) power-output equivalent or (*c*) normalized height-change equivalent, during ontogeny, in female common snapping turtles (*n* = 33). All three variables are positively correlated to body mass. Simple linear regressions were used to produce best-fit lines through the data. Height-change equivalent was normalized to carapace width. The ages in years of snapping turtles used in this study are indicated by closed circles, for 1.5 y (*n* = 26); open circles, for 4.5 y (*n* = 4); and half-closed circles, for 5.5 y (*n* = 3), as shown in the figure legend. Abbreviations: gram, g; body mass, $M_b$; kinetic-energy equivalent, KEE; power-output equivalent, PE; height-change equivalent, ΔHE. (Online version in colour.)

table S1). Furthermore, derived measures of sphericity and flatness were calculated and, unsurprisingly, given the likely geometric scaling of the shell shape, had no dependency on body mass, although individual values did show moderate variability (figure 2*b*; electronic supplementary material, figure S1). Neck length also fit the log–log model well ($R^2 = 0.895$); however, the 95% CI range (0.246–0.316) suggests that scaling is anisometric, with larger animals having shorter necks than would be predicted by geometric scaling (figure 2*c*, table 1; electronic supplementary material, table S1). The scatter might reflect greater ontogenetic variability or simply greater measurement uncertainty for this parameter.

## (b) Self-righting dynamics

To analyse self-righting performance, we plotted the log of self-righting time (defined as the duration of time in which the neck is applying force via the head) against the log of body mass (figure 2*d*). The OLS regression was significant (slope = 0.357, 95% CIs = 0.173–0.54; table 1) and the 95% CIs overlap the 0.5 value for the exponent predicted by our model based on geometric similarity. Smaller turtles self-righted proportionally faster (table 1; electronic supplementary material, table S1), but the duration of time it took for a turtle to place its head on the ground to initiate the self-righting manoeuvre (the pre-neck latency time) and the duration of time to complete the self-righting manoeuvre when the turtle no longer used the neck to flip over (the post-neck latency time) did not differ between the age groups (electronic supplementary material, figure S2).

We also plotted the log of KEE against the log of body mass (figure 3*a*) and found a modest effect, since the OLS slope is 1.548 (95% CIs = 1.341–1.755; table 1; electronic supplementary material, table S1), which does not overlap the 1.333 predicted by geometric similarity, suggesting the energy expended by a larger turtle is increasing more rapidly than our model would predict. To investigate the interaction between self-righting time and energetics, we plotted the log of mean of PE against the log of body

mass (figure 3*b*). We found an OLS slope of 1.191 (95% CIs = 0.961–1.422; table 1; electronic supplementary material, S1), which is higher than the 0.833 predicted by our model, indicating that the larger turtles are using higher power output to self-right than would be predicted by geometric scaling. To further illustrate how much more energetically expensive it is, for larger turtles, we calculated the shell width normalized height-change equivalent as a fraction of carapace width (figure 3*c*). This value should be unchanged with body mass, but, in fact, increases as the animals get larger.

## 4. Discussion

In most species, juveniles are more susceptible to mortality and often must avoid the same predators as adult conspecifics [26]. Natural selection tends to counteract this higher mortality, often by favouring improved locomotor performance through relatively longer limbs, faster muscle contractile velocities, and other physio-morphological changes that favour faster speeds and higher accelerations [26]. In the present study, we show that a simple geometric model, based on body mass, predicts shell shape and self-righting time when neck force is applied, in *C. serpentina*. However, we also show that energy and power outputs are greater during the self-righting process than would be predicted by our model (figure 3*a*,*b*, table 1; electronic supplementary material, table S1). The disproportionate increase in energetic cost is clearly shown by the height-change equivalent (figure 3*c*). Furthermore, younger turtles have disproportionally longer necks, which could be part of the reason they have lower-than-expected power outputs for their body size. Although our model does not predict the total time taken to self-right, smaller turtles complete the self-righting process faster than larger turtles in absolute terms and apply force via their neck for a shorter time (electronic supplementary material, figure S2, table S1). Considering that self-righting is a common locomotor behaviour exhibited by turtles [22], these scaling relationships and differences in self-righting might be widespread among other testudine species, to assist younger individuals in avoiding a

vulnerability that contributes to the high predation they face in nature [27].

## (a) Interaction between shell shape and self-righting effort

Our analyses demonstrated that shell shape in *C. serpentina* does not deviate from geometric scaling throughout ontogeny (figure 2c), and therefore cannot be associated with the changes seen in self-righting energy and power (figure 3a,b). In this respect *C. serpentina* appear to be different from some testudine species, in which juveniles inhabit a different micro-environment, which can drive morphological and biomechanical adaptations between life stages [28,29]. Accordingly, there are no morphological traits of the shell that would ameliorate the increased difficulty of self-righting as the turtles grow and age. Indeed, self-righting time increases with body mass, as predicted (figure 2d). Since the various risks of being inverted reduce with increased body size, this would support the idea that the evolutionary pressure is primarily on smaller turtles, considering there is no evidence of adaptations to reduce self-righting times in the larger animals. Our results parallel a study on Hermann's tortoises (*Testudo hermanni*), which shows that immature individuals self-right faster and with a higher probability of success than sexually mature adults [30]. Immature tortoises also display more anti-predatory behaviours, like boldness, and spend less time hiding in their shells during simulated predatory attacks, because their shells are weaker than adults [30]. Our data show that larger snapping turtles spend a longer time and disproportionately higher energy on self-righting when using the neck (even more than required by geometric similarity), suggesting that there are adaptations in juveniles for faster self-righting and to reduce its associated costs.

Although shell shape indices (sphericity and flatness) were not associated with changes in self-righting time in *C. serpentina* (figure 2b), they are good predictors of interspecific differences in self-righting in Testudines [20,23–25]. When comparing snapping turtles with two freshwater turtle species (that also use their necks to self-right), higher SIs are associated with faster self-righting time. The snapping turtle has the most domed shell (average SI = 0.758 ± 0.003) and self-rights fastest, followed by the red-eared slider (*Trachemys scripta elegans*; SI = 0.7 ± 0.01) [25], and then the Spanish terrapin (*Mauremys leprosa*; SI = 0.64 ± 0.004) [25]. These intraspecies differences might persist throughout life, given that shell sphericity does not vary after the hatchling life-stage, as found in *C. serpentina* (this study and [31]) and in *T. scripta* [25,32]. However, there are also instances of intraspecific differences in shell shape that are driven by habitat or sexual selection. For example, rainforest-dwelling scorpion mud turtles (*Kinosternon scorpioides*) have shorter shells than conspecifics living in dry forests [33], which are better for hiding, but would presumably hinder self-righting [20], and inverted male angulate tortoises (*Chersina angulata*), when battling other males for access to females, will self-right faster if they have a wider carapace [34]. Given the wide distribution of snapping turtles in North America [15], it would be interesting in further studies to determine whether there are geographical or sex differences in shell morphology that influence self-righting biomechanics.

## (b) Ontogeny and the scaling relationships of self-righting

In agreement with our hypothesis that self-righting would be completed faster in smaller, compared to larger individuals, we demonstrated that the youngest turtles self-right about twice as fast as the older cohorts, when neck force is applied. This is in line with the predictions of our model. However, the energetic effort is considerably lower for smaller turtles and the only morphological measure that does not scale geometrically is neck length, which is disproportionally longer in smaller turtles. The neck, in this case, can be considered an extra limb and the disproportionally longer necks of younger snapping turtles agree with anisometric scaling trajectories seen for limbs in other tetrapods. For example, allometric growth of bird wings [35] and shark caudal fins [36] have been interpreted as enabling juveniles to move with greater speed or agility than adults. Moreover, like other turtle species, snapping turtles rapidly project their necks to hunt, and neck length is primarily driven by prey-capture dynamics [37]. Because younger turtles are predominantly carnivorous, rather than omnivorous (like older turtles) [38], their disproportionately longer necks, would also be more efficient for seizing moving prey [37]. Thus, in younger snapping turtles, a relatively longer neck can serve at least two important functions: capturing prey more effectively and facilitating more energetically efficient self-righting.

To examine how a disproportionately shorter neck in larger turtles' affects self-righting effort, given that shell shape does not change, would require an investigation of the ontogenetic changes in neck musculature. Indeed, our findings of negative allometric neck growth and mass-specific neck force being independent of body mass (table 1; electronic supplementary material, S1) also fit the general pattern of growth in snapping turtles. During ontogeny, head size changes with negative allometry, whereas bite force scales isometrically, relative to carapace length [38]. Such scaling patterns suggest that the size, strength or physiology of the jaw muscles change throughout ontogeny, to preserve bite performance, despite a progressively smaller head [38]. Similar changes to the neck muscles might also occur during ontogeny, in snapping turtles. However, this remains to be determined.

## 5. Perspectives

In the present study, we have demonstrated that increasing body size during ontogeny increases self-righting times, as well as the accompanying biomechanical costs, and there is a reduction in the relative length of the neck, in snapping turtles. A young turtle's superior self-righting ability would be beneficial, as it would allow it to avoid the perils of being inverted as they traverse a landscape. Considering that *C. serpentina*, as well as other testudine species, possess shells optimized for the environment that they inhabit [2,3,5,33], intraspecific self-righting times and its biomechanical correlates could vary substantially, depending on geography. Therefore, future studies should investigate how the substrate from these different landscapes affects self-righting. Moreover, as interspecific differences in self-righting effort in Testudines is also influenced by shell shape, it is likely that it is also altered by shell rigidity. For example, common snapping turtles and spiny softshell turtles (*Apalone spinifera*) can

live in the same environments, but the latter have more flexible and smoother shells, and prefer to stay in water than on land. Thus, it would be worthwhile to investigate the trade-offs in self-righting ability of species that have flexible shells or spend most of their time in water. Lastly, because larger turtles use disproportionately more energy during self-righting, it begs the question of where this extra energy goes. Since the turtles do not leave the ground and the shell does not alter in shape, it must mean that the extra energy is likely converted into unnecessary body movement, or lost due to increased rolling resistance. Given the diversity and abundance of testudine species worldwide, as well as their vulnerability to anthropogenic and climate-change stressors, it is surprising how little we still know about some of their most basic biomechanical attributes that are associated with important survival behaviours, such as self-righting.

**Ethics.** Turtle husbandry and experimental procedures were carried out in accordance with an animal-care protocol (no. 11-007), approved by the University of North Texas Institutional Animal Care and Use Committee.

**Data accessibility.** The data associated with this study are available from the Dryad Digital Repository (https://doi.org/10.5061/dryad.mpg4f4qz5 [39]).

**Authors' contributions.** Conceptualization: J.R.C.; methodology: W.I.S. and J.R.C.; formal analysis: I.M.R. and K.A.R.R.; investigation: I.M.R., K.A.R.R., W.I.S., D.A.C. and J.R.C.; writing—original draft: I.M.R.; writing—review and editing: I.M.R, K.A.R.R., W.I.S., D.A.C. and J.R.C.; supervision: J.R.C.; project administration: J.R.C.; funding acquisition: J.R.C., W.I.S., K.A.R.R. and D.A.C.

**Competing interests.** The authors declare no competing interests.

**Funding.** This research was supported by the Leverhulme Trust (grant no. RPG-2019-104).

**Acknowledgement.** The authors would like to thank Janna Crossley for assistance with turtle training and husbandry.

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
