## [Peer Review File · Proceedings of the Royal Society B: Biological Sciences]

Review History

RSPB-2020-2965.R0 (Original submission)

Review form: Reviewer 1

Recommendation

Major revision is needed (please make suggestions in comments)

Scientific importance: Is the manuscript an original and important contribution to its field?

Good

General interest: Is the paper of sufficient general interest?

Good

Quality of the paper: Is the overall quality of the paper suitable?

Good

Is the length of the paper justified?

Yes

Should the paper be seen by a specialist statistical reviewer?

Yes

Do you have any concerns about statistical analyses in this paper? If so, please specify them explicitly in your report.

No

It is a condition of publication that authors make their supporting data, code and materials available - either as supplementary material or hosted in an external repository. Please rate, if applicable, the supporting data on the following criteria.

Is it accessible?

No

Is it clear?

N/A

Is it adequate?

N/A

Do you have any ethical concerns with this paper?

No

Comments to the Author

Table 2 -the raw data file- is missing from the main text and the supplement, and as such, the results could not be checked in detail. In addition, I suggest to add some references to the introduction and expand a bit on the Discussion, such as the transferability of the present results to other turtle species. I have marked all comments directly on the PDF (Appendix A).

Review form: Reviewer 2

Recommendation

Accept with minor revision (please list in comments)

Scientific importance: Is the manuscript an original and important contribution to its field?

Excellent

General interest: Is the paper of sufficient general interest?

Good

Quality of the paper: Is the overall quality of the paper suitable?

Excellent

Is the length of the paper justified?

Yes

Should the paper be seen by a specialist statistical reviewer?

No

Do you have any concerns about statistical analyses in this paper? If so, please specify them explicitly in your report.

No

It is a condition of publication that authors make their supporting data, code and materials available - either as supplementary material or hosted in an external repository. Please rate, if applicable, the supporting data on the following criteria.

Is it accessible?

No

Is it clear?

Yes

Is it adequate?

No

Do you have any ethical concerns with this paper?

No

Comments to the Author

This brief manuscript explores how carapace and neck size and measures self-righting energetics alter through ontogeny in a species of snapping turtle. I found this paper a pleasure to read and an interesting contribution to our understanding of turtle functional morphology and biomechanics. I have a few minor concerns that I outline below, but I think it is likely that they can be easily addressed to improve the manuscript for publication.

Detailed comments:

- Line 100 – “the biomechanics of self-righting neck force” is an odd phrase and I am puzzled at this stage what it actually means. I recognize that this is just a preview to bridge into the Methods Section, but a rewording to clarify what you mean here would be helpful.
- Lines 109-110 – be consistent with sample size abbreviations (capital vs lower-case).
- Line 110 – There is no Table 2 that I can find.
- Lines 128-130 – What is the spatial resolution of the pressure pad? In Figure 1E it appears that you are precisely localizing forces generated in different parts of the body and neck and in Figure 1F it appears that you are calculating precise proportions attributable to the neck relative to the body. What relative effect do forces generated closer to the neck have on the accuracy of these calculations compared to forces generated further from the neck?
- Lines 233-234 – Juveniles do not always live in identical environments as adults of the same species – see many lizard groups. In fact, the shift in habitat and microhabitat usage through development has been suggested to be driven in part by biomechanical and/or morphological differences between life stages. A bit mention of the complexity of this issue across diverse vertebrate groups and an explanation of how turtles are different in this respect would enrich your discussion here.
- Line 246 – “is” should be “in”
- Line 260-262 – Could you clarify this statement? Why does higher cost in adults imply an absence in evolutionary pressure in juveniles to reduce the cost?
- Line 282 – I do not think this is a fair statement given the amount of variation you have in your youngest cohort.
- Figure 1 – Panel E is confusing and the relevant portion of the caption does not help me understand what is being shown here. The panel does not appear to correspond to any of panels A-D and it is unclear when and where the forces are being generated during the behaviour. I highly suggest clarifying it perhaps by breaking it down into 4 panels and superimposing forces generated in each of the four stages illustrated as in panels A-D.
- Figure 1 caption – The last sentence refers to post-neck latency times in Figure 3B, but Figure 3B does not show anything related to this that I can see. It may also be helpful to briefly explain what pre- and post-neck latency time is in this caption. Also, what is the difference between absolute total force and total force in panel F?
- Figure 2 caption – define t=time abbreviation
- Figure 3 caption – power output equivalent should be PE, not P
- Supplementary table S1 – What is your justification for doing one-tailed tests? As you do not discuss these methods in the paper it needs to be justified in the supplementary material. Also, what is the purpose of comparing the turtles by age category when age is clearly an imprecise predictor of age given the large variation in size in your youngest category? This

seems an unnecessary and misleading generalization to me. How did you deal with the large difference in variation between your age categories?

- Supplementary material should contain all raw data but it does not. Please include raw data somewhere.

Decision letter (RSPB-2020-2965.R0)

05-Jan-2021

Dear Dr Codd:

I am writing to inform you that your manuscript RSPB-2020-2965 entitled "TURNING TURTLE: SCALING RELATIONSHIPS AND SELF-RIGHTING ABILITY IN CHELYDRA SERPENTINA" has, in its current form, been rejected for publication in Proceedings B.

This action has been taken on the advice of referees, who have recommended that substantial revisions are necessary. With this in mind we would be happy to consider a resubmission, provided the comments of the referees are fully addressed. However please note that this is not a provisional acceptance. The reviewers and AE are supportive but substantial revisions are needed.

Please note that this decision may (or may not) have taken into account confidential comments.

In your revision process, please take a second look at how open your science is; our policy is that ***ALL*** (maximally inclusive) data involved with the study should be made openly accessible, fully enabling re-use, replication and transparency-- see:

<https://royalsociety.org/journals/ethics-policies/data-sharing-mining/>

Insufficient sharing of data can delay or even cause rejection of a paper.

Full data and code/scripts to enable reuse/replication/repurposing are what this policy intends.

Note that Table 2, presumably including all the raw measurements, is missing. As such the results and graphs cannot be adequately assessed.

Sincerely,
Dr John Hutchinson, Editor
mailto: proceedingsb@royalsociety.org

Associate Editor
Board Member: 1
Comments to Author:

Thank you for the opportunity to read this paper. Herein the authors carry out a novel and interesting analysis of inversion and 'self-righting' in turtles. Specifically, they analyse the mechanics of how this self-righting is achieved, and its correlations with morphology (of the shell and neck), size and age. They recover a number of interesting correlations that culminate in finding that relatively longer necks in young turtles is causatively linked to higher mechanical costs in self-righting. I found this study very interesting to read and believe it delivers findings that will be of broad biological interest. This interest is largely shared by the two expert reviewers, although both have concerns that must be addressed before the manuscript can be considered further. In particular – all raw data must be provided.

Reviewer(s)' Comments to Author:

Referee: 1

Comments to the Author(s)

Table 2 -the raw data file- is missing from the main text and the supplement, and as such, the results could not be checked in detail. In addition, I suggest to add some references to the introduction and expand a bit on the Discussion, such as the transferability of the present results to other turtle species. I have marked all comments directly on the PDF.

Referee: 2

Comments to the Author(s)

This brief manuscript explores how carapace and neck size and measures self-righting energetics alter through ontogeny in a species of snapping turtle. I found this paper a pleasure to read and an interesting contribution to our understanding of turtle functional morphology and biomechanics. I have a few minor concerns that I outline below, but I think it is likely that they can be easily addressed to improve the manuscript for publication.

Detailed comments:

- Line 100 – “the biomechanics of self-righting neck force” is an odd phrase and I am puzzled at this stage what it actually means. I recognize that this is just a preview to bridge into the Methods Section, but a rewording to clarify what you mean here would be helpful.
- Lines 109-110 – be consistent with sample size abbreviations (capital vs lower-case).
- Line 110 – There is no Table 2 that I can find.
- Lines 128-130 – What is the spatial resolution of the pressure pad? In Figure 1E it appears that you are precisely localizing forces generated in different parts of the body and neck and in Figure 1F it appears that you are calculating precise proportions attributable to the neck relative to the body. What relative effect do forces generated closer to the neck have on the accuracy of these calculations compared to forces generated further from the neck?
- Lines 233-234 – Juveniles do not always live in identical environments as adults of the same species – see many lizard groups. In fact, the shift in habitat and microhabitat usage through development has been suggested to be driven in part by biomechanical and/or morphological differences between life stages. A bit mention of the complexity of this issue across diverse vertebrate groups and an explanation of how turtles are different in this respect would enrich your discussion here.
- Line 246 – “is” should be “in”

- Line 260-262 – Could you clarify this statement? Why does higher cost in adults imply an absence in evolutionary pressure in juveniles to reduce the cost?
- Line 282 – I do not think this is a fair statement given the amount of variation you have in your youngest cohort.
- Figure 1 – Panel E is confusing and the relevant portion of the caption does not help me understand what is being shown here. The panel does not appear to correspond to any of panels A-D and it is unclear when and where the forces are being generated during the behaviour. I highly suggest clarifying it perhaps by breaking it down into 4 panels and superimposing forces generated in each of the four stages illustrated as in panels A-D.
- Figure 1 caption – The last sentence refers to post-neck latency times in Figure 3B, but Figure 3B does not show anything related to this that I can see. It may also be helpful to briefly explain what pre- and post-neck latency time is in this caption. Also, what is the difference between absolute total force and total force in panel F?
- Figure 2 caption – define t=time abbreviation
- Figure 3 caption – power output equivalent should be PE, not P
- Supplementary table S1 – What is your justification for doing one-tailed tests? As you do not discuss these methods in the paper it needs to be justified in the supplementary material. Also, what is the purpose of comparing the turtles by age category when age is clearly an imprecise predictor of age given the large variation in size in your youngest category? This seems an unnecessary and misleading generalization to me. How did you deal with the large difference in variation between your age categories?
- Supplementary material should contain all raw data but it does not. Please include raw data somewhere.

Author's Response to Decision Letter for (RSPB-2020-2965.R0)

See Appendix B.

RSPB-2021-0213.R0

Review form: Reviewer 1

Recommendation

Accept as is

Scientific importance: Is the manuscript an original and important contribution to its field?

Good

General interest: Is the paper of sufficient general interest?

Good

Quality of the paper: Is the overall quality of the paper suitable?

Good

Is the length of the paper justified?

Yes

Should the paper be seen by a specialist statistical reviewer?

No

Do you have any concerns about statistical analyses in this paper? If so, please specify them explicitly in your report.

No

It is a condition of publication that authors make their supporting data, code and materials available - either as supplementary material or hosted in an external repository. Please rate, if applicable, the supporting data on the following criteria.

Is it accessible?

Yes

Is it clear?

Yes

Is it adequate?

Yes

Do you have any ethical concerns with this paper?

No

Comments to the Author

I think my previous comments and concerns have all been adequately addressed by the authors. A very minor last comment I would like to make is that the term Testudines, referring to crown-group turtles as a whole is correctly spelled with a capital T at the beginning, but in contrast the general reference to the group in the text as "testudine" should be with a small t (similar to Crocodylia and crocodylian or Squamata and squamate) instead.

Decision letter (RSPB-2021-0213.R0)

28-Jan-2021

Dear Dr Codd

I am pleased to inform you that your Review manuscript RSPB-2021-0213 entitled "TURNING TURTLE: SCALING RELATIONSHIPS AND SELF-RIGHTING ABILITY IN CHELYDRA SERPENTINA" has been accepted for publication in Proceedings B. Congratulations!! This is a nice paper.

The referee(s) do not recommend any further changes. Therefore, please proof-read your manuscript carefully and upload your final files for publication. Because the schedule for publication is very tight, it is a condition of publication that you submit the revised version of your manuscript within 7 days. If you do not think you will be able to meet this date please let me know immediately.

To upload your manuscript, log into <http://mc.manuscriptcentral.com/prsb> and enter your Author Centre, where you will find your manuscript title listed under "Manuscripts with Decisions." Under "Actions," click on "Create a Revision." Your manuscript number has been appended to denote a revision.

You will be unable to make your revisions on the originally submitted version of the manuscript. Instead, upload a new version through your Author Centre.

1) A text file of the manuscript (doc, txt, rtf or tex), including the references, tables (including captions) and figure captions. Please remove any tracked changes from the text before submission. PDF files are not an accepted format for the "Main Document".

2) A separate electronic file of each figure (tiff, EPS or print-quality PDF preferred). The format should be produced directly from original creation package, or original software format. Please note that PowerPoint files are not accepted.

3) Electronic supplementary material: this should be contained in a separate file from the main text and the file name should contain the author's name and journal name, e.g. `authorname_procb_ESM_figures.pdf`

All supplementary materials accompanying an accepted article will be treated as in their final form. They will be published alongside the paper on the journal website and posted on the online figshare repository. Files on figshare will be made available approximately one week before the accompanying article so that the supplementary material can be attributed a unique DOI. Please see: <https://royalsociety.org/journals/authors/author-guidelines/>

4) Data-Sharing and data citation

It is a condition of publication that data supporting your paper are made available. Data should be made available either in the electronic supplementary material or through an appropriate repository. Details of how to access data should be included in your paper. Please see <https://royalsociety.org/journals/ethics-policies/data-sharing-mining/> for more details.

<http://datadryad.org/submit?journalID=RSPB&manu=RSPB-2021-0213> which will take you to your unique entry in the Dryad repository.

Once again, thank you for submitting your manuscript to Proceedings B and I look forward to receiving your final version. If you have any questions at all, please do not hesitate to get in touch.

Sincerely,

Dr John Hutchinson

Associate Editor

Board Member

Comments to Author:

Thanks to the authors for engaging positively with the referees comments from the initial round of reviews, and for making their raw data available with this submission. I agree with the reviewers assessments that those initial comments have been addressed without any change or detracton from the findings or significance of the study.

Reviewer(s)' Comments to Author:

Referee: 1

Comments to the Author(s).

I think my previous comments and concerns have all been adequately addressed by the authors. A very minor last comment I would like to make is that the term Testudines, referring to crown-group turtles as a whole is correctly spelled with a capital T at the beginning, but in contrast the general reference to the group in the text as "testudine" should be with a small t (similar to Crocodylia and crocodylian or Squamata and squamate) instead.

Sincerely,

Proceedings B

Decision letter (RSPB-2021-0213.R1)

28-Jan-2021

Dear Dr Codd

I am pleased to inform you that your manuscript entitled "TURNING TURTLE: SCALING RELATIONSHIPS AND SELF-RIGHTING ABILITY IN CHELYDRA SERPENTINA" has been accepted for publication in Proceedings B.

Your article has been estimated as being 8 pages long. Our Production Office will be able to confirm the exact length at proof stage.

Open Access

Paper charges

Sincerely,
Proceedings B
[mailto: proceedingsb@royalsociety.org](mailto:proceedingsb@royalsociety.org)

Appendix A**PROCEEDINGS OF
THE ROYAL SOCIETY B**

BIOLOGICAL SCIENCES

**TURNING TURTLE: SCALING RELATIONSHIPS AND SELF-
RIGHTING ABILITY IN CHELYDRA SERPENTINA**

Journal:	Proceedings B
Manuscript ID	RSPB-2020-2965
Article Type:	Research
Date Submitted by the Author:	27-Nov-2020
Complete List of Authors:	Ruhr, Ilan; University of Manchester Faculty of Biology, Medicine and Health, Cardiovascular Sciences Rose, Kayleigh; Swansea University Sellers, William; University of Manchester, School of Earth and Environmental Sciences Crossley, Dane; University of North Texas, Biological Sciences Codd, Jonathan; University of Manchester
Subject:	Biomechanics < BIOLOGY, Evolution < BIOLOGY
Keywords:	chelonian, morphology, neck, allometry, ontogeny, biomechanics
Proceedings B category:	Morphology & Biomechanics

Author-supplied statements

Relevant information will appear here if provided.

Ethics

Does your article include research that required ethical approval or permits?:

This article does not present research with ethical considerations

Statement (if applicable):

CUST_IF_YES_ETHICS :No data available.

Data

It is a condition of publication that data, code and materials supporting your paper are made publicly available. Does your paper present new data?:

Yes

Statement (if applicable):

additional data is provided in the electronic supplementary material

Conflict of interest

I/We declare we have no competing interests

Statement (if applicable):

CUST_STATE_CONFLICT :No data available.

Authors' contributions

This paper has multiple authors and our individual contributions were as below

Statement (if applicable):

Conceptualization: J.R.C.; Methodology: W.I.S. and J.R.C.; Formal analysis: I.M.R. and K.A.R.R.; Investigation: I.M.R., K.A.R.R., W.I.S., D.A.C., and J.R.C.; Writing “ original draft: I.M.R.; writing “ review and editing: I.M.R, K.A.R.R., W.I.S., D.A.C., and J.R.C.; Supervision: J.R.C.; Project administration: J.R.C.; Funding acquisition: J.R.C., W.I.S., K.A.R.R., and D.A.C.

TURNING TURTLE: SCALING RELATIONSHIPS AND SELF-RIGHTING ABILITY IN CHELYDRA SERPENTINA

Ilan M. Ruhr¹, Kayleigh A. R. Rose², William I. Seller³, Dane A. Crossley, II⁴, & Jonathan R. Codd^{1*}

¹School of Biological Sciences, University of Manchester, UK; ²Department of Biosciences, Swansea University, Swansea, UK; ³Department of Earth and Environmental Sciences, University of Manchester, UK; ⁴Department of Biological Sciences, University of North Texas, USA

***CORRESPONDING AUTHOR**

jonathan.codd@manchester.ac.uk

KEYWORDS: chelonian, morphology, neck, allometry, ontogeny, biomechanics

33 ABSTRACT

[revised manuscript text omitted]
**Legrand, A., & Cambag, R.** 2001 Sexual dimorphism in steppe tortoises (*Testudo horsfieldii*): influence of
the environment and sexual selection on body shape and mobility. *Biological Journal of the Linnean*
*Society* **72**, 357-372. (DOI:10.1111/j.1095-8312.2001.tb01323.x).
- 11. **Burger, J.** 1976 Behavior of hatchling diamondback terrapins (*Malaclemys terrapin*) in the field. *Copeia*
**1976**, 742. (DOI:10.2307/1443457).
- 12. **Delmas, V., Baudry, E., Girondot, M., & Prevot-Julliard, A. C.** 2007 The righting response as a fitness
index in freshwater turtles. *Biol J Linn Soc* **91**, 99-109. (DOI:10.1111/j.1095-8312.2007.00780.x).
- 13. **Iverson, J. B., Higgins, H., Sirulnik, A., & Griffiths, C.** 1997 Local and geographic variation in the
reproductive biology of the snapping turtle (*Chelydra serpentina*). *Herpetologica* **53**, 96-117.
- 14. **Congdon, J. D., Nagle, R. D., Dunham, A. E., Beck, C. W., Kinney, O. M., & Yeomans, S. R.** 1999 The
relationship of body size to survivorship of hatchling snapping turtles (*Chelydra serpentina*): an
evaluation of the "bigger is better" hypothesis. *Oecologia* **121**, 224-235. (DOI:10.1007/s004420050924).
- 15. **Sellers, W. I., Rose, K., Crossley, D. A., II, & Codd, J. R.** 2020 Inferring cost of transport from whole-body
kinematics in three sympatric turtle species with different locomotor habits. *Comp Biochem Physiol A*
*Mol Integr Physiol*, 110739. (DOI:10.1016/j.cbpa.2020.110739).
- 16. **Rivera, A. R. V., Rivera, G., & Blob, R. W.** 2004 Kinematics of the righting response in inverted turtles.
*Journal of Morphology* **260**, 274-342. (DOI:10.1002/jmor.10223).
- 17. **Rubin, A. M., Blob, R. W., & Mayerl, C. J.** 2018 Biomechanical factors influencing successful self-righting
in the pleurodire turtle, *Emydura subglobosa*. *J Exp Biol* **221**, jeb182642. (DOI:10.1242/jeb.182642).
- 18. **Domokos, G. & Varkonyi, P. L.** 2008 Geometry and self-righting of turtles. *Proc Biol Sci* **275**, 11-17.
(DOI:10.1098/rspb.2007.1188).
- 19. **Golubović, A., Bonnet, X., Djordjević, S., Djuracic, M., & Tomović, L.** 2013 Variations in righting
behaviour across Hermann's tortoise populations. *J Zool* **291**, 69-75. (DOI:10.1111/jzo.12047).
- 20. **Ashe, V. M.** 1970 The righting reflex in turtles: A description and comparison. *Psychon Sci* **20**, 150-152.
(DOI:10.3758/BF03335647).
- 21. **Ana, G., Ljiljana, T., & Ana, I.** 2015 Geometry of self righting: the case of Hermann's tortoises. *Zool Anz*
**254**, 99-105. (DOI:10.1016/j.jcz.2014.12.003).
- 22. **Chiari, Y., van der Meijden, A., Caccone, A., Claude, J., & Gilles, B.** 2017 Self-righting potential and the
evolution of shell shape in Galapagos tortoises. *Sci Rep* **7**, 15828. (DOI:10.1038/s41598-017-15787-7).
- 23. **Polo-Cavia, N., López, P., & Martín, J.** 2012 Effects of body temperature on righting performance of
native and invasive freshwater turtles: consequences for competition. *Physiol Behav* **108**, 28-33.
(DOI:10.1016/j.physbeh.2012.10.002).
- 24. **Carrier, D. R.** 1996 Ontogenetic limits on locomotor performance. *Physiol Zool* **69**, 467-488.
(DOI:10.1086/physzool.69.3.30164211).

- 25. **Brooks, R. J., Brown, G. P., & Galbraith, D. A.** 1991 Effects of a sudden increase in natural mortality of
adults on a population of the common snapping turtle (*Chelydra serpentina*). *Can J Zool* **69**, 1314-1320.
(DOI:10.1139/z91-185).
- 26. **Golubović, A.** 2015 Ontogenetic shift of antipredator behaviour in Hermann's tortoises. *Behav Ecol*
*Sociobiol* **69**, 1201-1208. (DOI:10.1007/s00265-015-1934-9).
- 27. **Mosimann, J. E. & Bider, J. R.** 1960 Variation, sexual dimorphism, and maturity in a Quebec population
of the common snapping turtle, *Chelydra Serpentina*. *Can J Zool* **38**, 19-38. (DOI:10.1139/z60-003).
- 28. **Fish, J. F. & Stayton, C. T.** 2014 Morphological and mechanical changes in juvenile red-eared slider turtle
(*Trachemys scripta elegans*) shells during ontogeny. *J Morphol* **275**, 391-397. (DOI:10.1002/jmor.20222).
- 29. **Acuna-Mesen, R. A.** 1994 Morphometric variation and ecologic characteristics of the habitat of the
*Kinosternon scorpioides* turtle in Costa Rica (Chelonia, Kinosternidae). *Rev Bras Biol* **54**, 537-547.
- 30. **Mann, G. K. H., O'Riain, M. J., & Hofmeyr, M. D.** 2006 Shaping up to fight: sexual selection influences
body shape and size in the fighting tortoise (*Chersina angulata*). *J Zool* **269**, 373-379.
(DOI:10.1111/j.1469-7998.2006.00079.x).
- 31. **Dudley, R.** 2002 Mechanisms and implications of animal flight maneuverability. *Integr Comp Biol* **42**, 135-
140. (DOI:10.1093/icb/42.1.135).
- 32. **Irschick, D. J. & Hammerschlag, N.** 2015 Morphological scaling of body form in four shark species
differing in ecology and life history. *Biol J Linn Soc* **114**, 126-135. (DOI:10.1111/bij.12404).
- 33. **Pritchard, P. C. H.** 1984 Piscivory in turtles, and evolution of the long-necked *Chelidae*. *Zool Soc Lon* **52**,
87-110.
- 34. **Herrel, A. & O'Reilly J, C.** 2006 Ontogenetic scaling of bite force in lizards and turtles. *Physiol Biochem*
*Zool* **79**, 31-42. (DOI:10.1086/498193).

**ACKNOWLEDGEMENTS**

**FUNDING**

This research was supported by The Leverhulme Trust (RPG-2019-104).

**GENERAL**

The authors would like to thank Janna Crossley for assistance with turtle training and husbandry.

**AUTHOR CONTRIBUTIONS**

Conceptualization: J.R.C.; Methodology: W.I.S. and J.R.C.; Formal analysis: I.M.R. and K.A.R.R.; Investigation:
I.M.R., K.A.R.R., W.I.S., D.A.C., and J.R.C.; Writing – original draft: I.M.R.; writing – review and editing: I.M.R.,
415 K.A.R.R., W.I.S., D.A.C., and J.R.C.; Supervision: J.R.C.; Project administration: J.R.C.; Funding acquisition: J.R.C.,
416 W.I.S., K.A.R.R., and D.A.C.

**COMPETING INTERESTS**

The authors declare no competing interests.

**DATA ACCESSIBILITY**

The dataset associated with this study is available from the Dryad Digital Repository.

**TABLES**

Table 1. Scaling relationships between morphometric/biomechanical variables and body mass or between morphometrics and maximum neck force. The regression slope indicates proportional change in variable size with increasing body mass, and 95% confidence intervals (CIs) are shown (N = 33). Measured slopes in agreement (using 95% confidence intervals) with predicted slopes from our geometric model are indicated by a check mark (✓) and measured slopes lesser or greater than model predictions are indicated by negative signs (–) and positive signs (+), respectively.

Dependent variable	Independent variable	Slope predicted by geometric model	Measured slope	In agreement with model prediction?	Lower 95% CI	Upper 95% CI	R ²	P-value
Carapace length (mm)	Body mass (kg)	0.33	0.323	✓	0.31	0.336	0.988	≤ 0.001
Carapace width (mm)	Body mass (kg)	0.33	0.336	✓	0.318	0.354	0.979	≤ 0.001
Carapace height (mm)	Body mass (kg)	0.33	0.31	✓	0.295	0.335	0.984	≤ 0.001
Neck length (mm)	Body mass (kg)	0.33	0.281	–	0.246	0.316	0.895	≤ 0.001
Self-righting time (s)	Body mass (kg)	0.5	0.357	✓	0.173	0.54	0.337	0.004
Neck force (N)	Body mass (kg)	0.67	1.040	+	0.831	1.249	0.699	≤ 0.001
Kinetic energy equivalent (J)	Body mass (kg)	1.33	1.542	+	1.335	1.749	0.882	≤ 0.001
Power output (W)	Body mass (kg)	0.83	1.186	+	0.956	1.415	0.781	≤ 0.001
Neck force (N)	Neck length (mm)	2	2.799	✓	2.027	3.570	0.639	≤ 0.001
	Carapace length (mm)	2	2.644	✓	1.985	3.304	0.683	≤ 0.001
	Carapace width (mm)	2	2.525	✓	1.891	3.159	0.68	≤ 0.001
	Carapace height (mm)	2	2.721	✓	2.021	3.422	0.67	≤ 0.001

425

Figure 1. Representative images showing a sequence of the progressive stages in a self-righting snapping turtle. (A) initial contact of ❶ the neck to the ground; (B) ❷ extension of the neck, which ❸ begins lifting the shell off the ground; (C) ❹ full extension, ❺ twisting of the neck, and ❻ rotation of the shell about its axis; and (D) completion of the self-righting maneuver, with all four limbs contacting the ground. White lines with arrows represent the direction of motion. The biomechanical effort involved for different body areas is depicted as (E) a superimposed image of maximum-force peaks, produced by a turtle as it self-rights. The relative magnitude of the force (F magnitude) production is indicated by colours, as shown in the panel legend. (F) Representative traces of absolute total force (F_{total} ; black line) in Newtons (N) produced by a snapping turtle during one self-righting maneuver and recorded by a force plate (indicated in the panel legend). A pressure pad recorded the contribution of neck force (F_{neck} ; blue line) and total force (red line) exerted by the turtle during self-righting (indicated in the panel legend to the left). The sequence of the self-righting maneuver labelled from A to D, to match the corresponding panels are indicated over the blue F_{neck} line. The red, blue, and purple rectangular sections found under the x-axis correspond to the pre-neck latency, self-righting, and post-neck latency times that are shown in Fig. 3B, for reference.

75x41mm (600 x 600 DPI)

Figure 2. Relationships with body mass between (A) carapace length (red triangles), carapace width (blue triangles), and shell height (green triangles); (B) shell shape (sphericity and flatness indices); (C) neck length, during ontogeny; and (D) self-righting time, during ontogeny, in female common snapping turtles ($N = 33$). All morphometric parameters positively correlated with body mass. Carapace length, carapace width, and shell height followed isometric growth, whereas neck length followed a negative allometric pattern. Self-righting time, during which a snapping turtle uses its head to flip over, is positively correlated with body mass and follows isometric scaling. Simple linear regressions were used to produce best-fit lines through the data. The ages (in years) of snapping turtles used in this study are indicated by closed triangles, for 1.5 y ($n = 26$); open triangles, for 4.5 y ($n = 4$); and half-closed triangles, for 5.5 y ($n = 3$), as shown in the figure legend. Abbreviations: body mass, M_b ; grams, g; length, L; width, W; height, H.

42x41mm (600 x 600 DPI)

Figure 3. Relationship between body mass and (A) kinetic-energy equivalent, (B) power-output equivalent, or (C) normalized height-change equivalent, during ontogeny, in female common snapping turtles ($N = 33$). All three variables are positively correlated to body mass. Simple linear regressions were used to produce best-fit lines through the data. Height-change equivalent was normalized to carapace width. The ages in years of snapping turtles used in this study are indicated by closed circles, for 1.5 y ($n = 26$); open circles, for 4.5 y ($n = 4$); and half-closed circles, for 5.5 y ($n = 3$), as shown in the figure legend. Abbreviations: gram, g; body mass, M_b ; kinetic-energy equivalent, KEE; power-output equivalent, P; height-change equivalent, ΔHE .

61x22mm (600 x 600 DPI)

Appendix B

TURNING TURTLE: SCALING RELATIONSHIPS AND SELF-RIGHTING ABILITY IN CHELYDRA SERPENTINA

Ilan M. Ruhr¹, Kayleigh A. R. Rose², William I. Seller³, Dane A. Crossley, II⁴, & Jonathan R. Codd^{1*}

Handling Editor Comments: Thank you for the opportunity to read this paper. Herein the authors carry out a novel and interesting analysis of inversion and 'self-righting' in turtles. Specifically, they analyse the mechanics of how this self-righting is achieved, and its correlations with morphology (of the shell and neck), size and age. They recover a number of interesting correlations that culminate in finding that relatively longer necks in young turtles is causatively linked to higher mechanical costs in self-righting. I found this study very interesting to read and believe it delivers findings that will be of broad biological interest. This interest is largely shared by the two expert reviewers, although both have concerns that must be addressed before the manuscript can be considered further. In particular – all raw data must be provided.

RESPONSE: we thank for editor for the positive assessment of our manuscript. We apologize for not including a link to the raw data this was an oversight on our part and we have now included a link to the DRYAD data repository and a detailed description of how these data were handled to generate the results included in our manuscript. We have addressed the remaining minor comments below:

Responses to Referee 1: from comments annotated on the PDF

Line 34: The term Chelonian has been changed to Testudine throughout as suggested.

Line 51-52: We have altered the sentence to read; "For example, generally,...." And we have added two references as suggested to better support the claims related to protection.

Line 58: Sentence has been re-written "because the spine is fused with the underside of the dermal plates to form a hard shell"

Line 64: as the suggested reference relates more to habitat determination we have chosen not to reference it as it does not add to the point we are making.

Line 109-110: Yes we agree that these are young turtles. We apologize for the error in omitting the raw data files this was an oversight, we have now included these through a DRYAD links in the manuscript. Table 2 should be a reference to the DRYAD link and has been corrected.

Line 185: Yes this should be KEE – changed as suggested.

Line 204: changed for clarity to read "shell shape, as defined in the current study, ..."

Line 234: We agree with the referee and are happy that our statement is correct.

Line 246: change to in as suggested.

Line 247: we agree that it would be very interesting to determine the risk of inverting in the wild for this (and other) species but these data don't exist yet. We do know they do invert and the point of our paper was to assess this behaviour throughout ontogeny. We also agree that it would be fascinating to conduct these studies on other species so that we could get an idea on how transferable our results are – work we plan to do in the future.

Line 250: change to '*C.serpentina*' to clarify that we are referring to the species we studied in this paper and not all snapping turtles.

Line 271: changed to "as the turtles..." for clarity.

Line: 277: change to read ".....interesting in further studies..." for clarity.

Line 298: ';' replaced with a full stop as suggested.

Line 309: we agree and this point has been made previously.

Line 331: changed to "*C. serpentine*" for clarity.

Responses to Referee 2:

Line 100 – “the biomechanics of self-righting neck force” is an odd phrase and I am puzzled at this stage what it actually means. I recognize that this is just a preview to bridge into the Methods Section, but a rewording to clarify what you mean here would be helpful.

RESPONSE: we agree that this is badly worded and have changed it by removing ‘the biomechanics...’

Lines 109-110 – be consistent with sample size abbreviations (capital vs lower-case).

RESPONSE: changed as suggested to the small ‘n’.

Line 110 – There is no Table 2 that I can find.

RESPONSE: this was an error and has been corrected to refer to the ESM & DRYAD data files.

Lines 128-130 – What is the spatial resolution of the pressure pad? In Figure 1E it appears that you are precisely localizing forces generated in different parts of the body and neck and in Figure 1F it appears that you are calculating precise proportions attributable to the neck relative to the body. What relative effect do forces generated closer to the neck have on the accuracy of these calculations compared to forces generated further from the neck?

RESPONSE: The spatial resolution of the pressure pad is in 5mm squares. They measure pressure but since we know the area, they also give the mean vertical force over the 5mm square. Each 'sense' is independent so they do measure the spatial distribution of the reaction force. Figure 1E just shows a snapshot of force distribution during self-righting (which has been clarified in response to another referee comment below) we include this just to show a representative trace of what the force distribution looks like at a specific time in the self-righting event. As all forces are measured spatially we can identify specifically the forces being applied through the neck and use these for analyses. Because we can isolate the force from the neck there is no influence of the forces from other parts of the body.

Lines 233-234 – Juveniles do not always live in identical environments as adults of the same species – see many lizard groups. In fact, the shift in habitat and microhabitat usage through development has been suggested to be driven in part by biomechanical and/or morphological differences between life stages. A bit mention of the complexity of this issue across diverse vertebrate groups and an explanation of how turtles are different in this respect would enrich your discussion here.

RESPONSE: we agree that this is a little clumsy in the way it is worded we have corrected this by altering the text removing the reference to competing in identical environments: “...juveniles are more susceptible to mortality and often must avoid the same predators as adult...” and added a sentence to broaden the discussion: “In this respect *C. serpentina* appear to be different from some Testudine species in which juveniles inhabit a different micro environment which can drive morphological and biomechanical adaptations between life stages (Arthur et al 2008; Lamont et al 2015)”.

Line 246 – “is” should be “in”

RESPONSE: corrected as suggested.

Line 260-262 – Could you clarify this statement? Why does higher cost in adults imply an absence in evolutionary pressure in juveniles to reduce the cost?

RESPONSE: this sentence has been re-written for clarity and now reads “...suggesting that there are adaptations in juveniles for faster self-righting and to reduce its associated costs”.

Line 282 – I do not think this is a fair statement given the amount of variation you have in your youngest cohort.

RESPONSE: our data show clearly that the youngest turtles do right on average about twice as fast as the old larger turtles – our statistical analysis considers any variation in the groups.

Figure 1 – Panel E is confusing and the relevant portion of the caption does not help me understand what is being shown here. The panel does not appear to correspond to any of panels A-D and it is unclear when and where the forces are being generated during the behaviour. I highly suggest clarifying it perhaps by breaking it down into 4 panels and superimposing forces generated in each of the four stages illustrated as in panels A-D.

RESPONSE: we agree this was confusing and have tidied up the figure and rewritten the legend, we have swapped out the panel B figure so that the turtle image now matches the representative force distribution image in panel E, we have also in the legend elaborated on what is being shown to take away any misunderstandings; it now reads: “a representative image showing the integrated force peaks during the self-righting manoeuvre depicted in Panel B, highlighting the dynamic distribution of forces across the turtle and the concentration of force application via the neck. The relative magnitude of the force (F magnitude) production is indicated by colours, as shown in the panel legend”.

The figure panel is included as a simple visual aid to give an example of the force application during self-righting.

Figure 1 caption – The last sentence refers to post-neck latency times in Figure 3B, but Figure 3B does not show anything related to this that I can see. It may also be helpful to briefly explain what pre- and post-neck latency time is in this caption. Also, what is the difference between absolute total force and total force in panel F?

RESPONSE: The reference to Figure 3B was an error and this has been corrected to reference Figure S2 in the ESM. The differences between the two-force measures is already explained in the figure legend – absolute total force is the force measured by the force plate the total force is that measured by the pressure pad: “Representative traces of **absolute total force** (F_{total} ; black line) in Newtons (N) produced by a snapping turtle during one self-righting manoeuvre and recorded by a force plate (indicated in the panel legend). A pressure pad recorded the contribution of neck force (F_{neck} ; blue line) and **total force** (red line) exerted by the turtle during self-righting (indicated in the panel legend to the left).

Figure 2 caption – define t=time abbreviation.

RESPONSE: changed as suggested.

Figure 3 caption – power output equivalent should be PE, not P.

RESPONSE: changed as suggested.

Supplementary table S1 – What is your justification for doing one-tailed tests? As you do not discuss these methods in the paper it needs to be justified in the supplementary material. Also, what is the purpose of comparing the turtles by age category when age is clearly an imprecise predictor of age given the large variation in size in your youngest category? This seems an unnecessary and misleading generalization to me. How did you deal with the large difference in variation between your age categories?

RESPONSE: we use t-tests as these are the appropriate statistical test to examine significance in one direction of interest - in our case the factors that we are studying have a clearly demonstrated difference as size changes. We use age as a delimitator of turtle size – any variation in body size is considered with our analyses by using it as a factor.

Supplementary material should contain all raw data but it does not. Please include raw data somewhere.

RESPONSE: we agree – see earlier comments where this has been corrected.